# Absence of HDAC3 by Matrix Stiffness Promotes Chromatin Remodeling and Fibroblast Activation in Idiopathic Pulmonary Fibrosis

**DOI:** 10.3390/cells12071020

**Published:** 2023-03-27

**Authors:** Fernanda Toscano-Marquez, Yair Romero, Marco Espina-Ordoñez, José Cisneros

**Affiliations:** 1Laboratorio de Biopatología Pulmonar INER-Ciencias-UNAM, Departamento de Fibrosis Pulmonar, Instituto Nacional de Enfermedades Respiratorias Ismael Cosío Villegas (INER), Mexico City 14080, Mexico; 2Posgrado en Ciencias Biológicas, Unidad de Posgrado, Universidad Nacional Autónoma de México, Mexico City 04510, Mexico; 3Facultad de Ciencias, Universidad Nacional Autónoma de México, Mexico City 04510, Mexico

**Keywords:** IPF, histone deacetylase 3, epigenetics, matrix stiffness, polyacrylamide hydrogels

## Abstract

Idiopathic pulmonary fibrosis (IPF) is a chronic and fatal disease characterized by progressive and irreversible lung scarring associated with persistent activation of fibroblasts. Epigenetics could integrate diverse microenvironmental signals, such as stiffness, to direct persistent fibroblast activation. Histone modifications by deacetylases (HDAC) may play an essential role in the gene expression changes involved in the pathological remodeling of the lung. Particularly, HDAC3 is crucial for maintaining chromatin and regulating gene expression, but little is known about its role in IPF. In the study, control and IPF-derived fibroblasts were used to determine the influence of HDAC3 on chromatin remodeling and gene expression associated with IPF signature. Additionally, the cells were grown on hydrogels to mimic the stiffness of a fibrotic lung. Our results showed a decreased HDAC3 in the nucleus of IPF fibroblasts, which correlates with changes in nucleus size and heterochromatin loss. The inhibition of HDAC3 with a pharmacological inhibitor causes hyperacetylation of H3K9 and provokes an increased expression of Col1A1, ACTA2, and p21. Comparable results were found in hydrogels, where matrix stiffness promotes the loss of nuclear HDAC3 and increases the profibrotic signature. Finally, latrunculin b was used to confirm that changes by stiffness depend on the mechanotransduction signals. Together, these results suggest that HDAC3 could be a link between epigenetic mechanisms and the fibrotic microenvironment.

## 1. Introduction

Idiopathic pulmonary fibrosis (IPF) is an age-related, progressive, chronic, and fatal disease within a few years, characterized by inadequate remodeling of the lung parenchyma. The pathogenesis of the disease has been proposed as epithelial-driven fibrosis because these cells are capable of fibroblast activation, responsible for the exacerbated secretion of extracellular matrix (ECM) components [1,2]. However, the mechanisms by which the profibrotic phenotype is established in fibroblasts still need to be fully understood.

Epigenetic mechanisms play a crucial role in integrating the stimuli from the microenvironment, leading to gene expression changes. In IPF, this epigenetic reprogramming occurs by the interaction of the cell–microenvironment, establishing a profibrotic feedback loop favoring their differentiation to myofibroblasts and the expression of genes involved in the pathological remodeling of the extracellular matrix [3,4,5,6]. Among the epigenetic mechanisms, acetylation and deacetylation of histones are strategic modifications that regulate accessibility to DNA [7]. The deacetylation process is carried out by histone deacetylases (HDACs), a family of enzymes divided into four classes (I–IV) that remove acetyl groups in histone tails to favor the interaction of DNA with nucleosomes and heterochromatin formation, preventing gene transcription [8]. Abnormal HDAC expression has been observed in fibrotic diseases, including pulmonary fibrosis [9,10,11].

HDAC3 belongs to Class I and has both nuclear and cytoplasmic localization. In the nucleus, its activity is non-redundant with the other deacetylases because it has specific substrates, among which are lysine 9 of histone 3 (H3K9ac) and lysine 5 (H4K5ac) and 12 from histone 4 (H4K12ac) [12,13,14]. HDAC3 regulates these marks by forming a multiprotein complex with the corepressors SMRT and N-CoR [15]. Studies reported in the literature regarding its activity demonstrated that deletion of this protein results in the global loss of heterochromatin and genomic instability that leads to the transcription of tumor genes [16]. Moreover, HDAC3 participates in the specific repression of genes associated with differentiation by binding gene regions to the nuclear lamina [16,17]. The role of HDAC3 in lung fibroblasts and its impact on the development and progression of IPF still remain unclear.

In the present study, we aimed to determine the role of HDAC3 in regulating fibroblast chromatin and profibrotic gene expression and its putative role in integrating the influence of matrix stiffness. We examined the HDAC3 localization in control and IPF lung fibroblasts and the impact of its inhibition on nuclear morphology, histone modifications, and profibrotic gene expression.

## 2. Materials and Methods

### 2.1. Cell Culture and Reagents

Primary lung fibroblasts derived from IPF patients (*n* = 3) previously obtained in our laboratory and controls purchased from Lonza were used. Fibroblasts were cultured at 37 °C in 5% CO_2_/95% air in Advanced DMEM/F-12 Gibco™ medium (Fisher Scientific, Hampton, NH, USA) supplemented with 5% FBS and 100 µg/mL of antimicrobial agent (Primocin; InvivoGen, San Diego, CA, USA). All fibroblasts were used between passages 5 and 9 and were seeded on plastic cell culture plates (6 and 12 wells) or in polyacrylamide hydrogels (HGs) of different stiffness. IPF and control fibroblasts at 80% of confluence were treated with either 10 μM of RGFP966 (Cayman Chemical, Ann Arbor, MI, USA) or DMSO vehicle control for 24 h. 

### 2.2. Immunofluorescence

Cells grown on coverslips were washed with PBS, fixed with p-formaldehyde 4% for 30 min at room temperature, permeabilized, and blocked with Triton x-100 0.1% in PBS with BSA 4% for 30 min at room temperature. Samples were incubated with primary antibodies overnight at 4 °C (HDAC3 1:50 (GeneTex, Irvine, CA, USA), H3K9ac 1:100 (SCBT), H3K9me2 1:100 (GeneTex), Lamin A/C 1:200 (GeneTex) and washed with PBS, followed by incubation with Alexa Fluor™ (Thermo Scientific, Waltham, MA, USA) secondary antibodies for 1 h at room temperature. F-actin was stained with Phalloidin Alexa Fluor™ 488 (Thermo Scientific) and nuclei with DAPI Fluorescent Stain (Thermo Scientific). Coverslips were examined and photographed under a confocal laser-scanning Olympus FluoView™ FV1000 microscope or under the fluorescence microscope Olympus IX81.

### 2.3. Immunoblotting

Cells were lysed in RIPA buffer (Thermo Scientific) or cytoplasmic and nuclear fractions were obtained with an NE-PER kit (Thermo Scientific). Proteins were quantified with bicinchoninic acid assay (Thermo Scientific). Samples were mixed in a 4X Laemmli buffer (BioRad, Hercules, CA, USA) with 2-mercaptoethanol and heated for 5 min at 95 °C. Then, 25 µg of protein was used for electrophoresis in a 10 or 12% polyacrylamide gel. Proteins were transferred into nitrocellulose membranes (BioRad), which were blocked for 1 h at room temperature using an Li-Cor blocking buffer (Licor, Lincoln, NE, USA). Primary antibodies for HDAC3 (1:750, GeneTex), H3K9ac (1:1000, SCBT, Dallas, TX, USA), H3K9me2 (1:1000, GeneTex), Lamin A/C 1:2000 (GeneTex), Lamin A 1:1000, GAPDH (1:3000, SCBT, Dallas, TX, USA), or H3 (1:1000, GeneTex) were incubated overnight, and then incubated with secondary antibodies (Li-Cor, Lincoln, NE, USA). Membranes were developed by fluorescence in the Odyssey scanner (Li-Cor, Lincoln, NE, USA). Quantification was performed using ImageJ software Version 1.53t.

### 2.4. Transmission Electron Microscopy

Lung fibroblasts were cultured in thermanox plastic in a 24-well plate. The electron microscopy technique was used, as previously described in [17]. Briefly, fixed samples were treated with osmium tetroxide and stained with uranyl acetate. Samples were dehydrated in ascending grades of ethanol and embedded in a resin Embed 812 kit (Electron Microscopy Sciences, Hatfield, PA, USA). Thin sections (70 nM) were analyzed with a Tecnai G2 Spirit transmission electron microscope (FEI Company, Hillsboro, OR, USA).

### 2.5. Quantitative PCR

The RNA was isolated using Trizol reagent (Invitrogen, Carlsbad, CA, USA) according to the manufacturer’s protocol. cDNA was synthesized using 1 ug of RNA using the Verso cDNA Synthesis Kit (Thermo Scientific), following the kit protocol. Quantitative PCR amplification was performed in a mixture containing 10 ng of cDNA, SYBR green (Thermo Fisher, Waltham, MA, USA), and primers designed on the platform PrimerBank. Thermocycler QuantStudio 6 (Applied Biosystems, Waltham, MA, USA) was used for qPCR experiments and 2^−∆Ct^ was used for relative quantification data analysis. The following primer sequences were used:

HPRT-R: GGCTTTGTATTTTGCTTTTCCA, HPRT-F: AGGACCCCACGAAGTGTT; COL1A1-R: CAGATCACGTCATCGCACAAC, COL1A1-F: GAGGGCCAAGACGAAGACATC; ACTA2-R: GCCATGTTCTATCGGGTACTTC, ACTA2-F: AAAAGACAGCTACGTGGGTGA; TGFβ1-R: GTGGGTTTCCACCATTAGCAC, TGFβ1-F: GGCCAGATCCTGTCCAAGC; CDKN1-R: GGGTCTGTAGTAGAACTCGGG, CDKN1-F: ATCACAAACCCCTAGAGGGCA; PCNA-R: CAGCGGTAGGTGTCGAAGC, PCNA-F: CCTGCTGGGATATTAGCTCCA; MKI67-R: CAGACCCATTTACTTGTGTTGGA, MKI67-F: ACGCCTGGTTACTATCAAAAGG; PLK1-R: GGCTGCGGTGAATGGATATTTC, PLK1-F: AAAGAGATCCCGGAGGTCCTA.

### 2.6. Polyacrylamide Hydrogels

Polyacrylamide hydrogels were made to mimic the stiffness of a healthy (1 kPa) and fibrotic (23 kPa) lung [18]. Briefly, the hydrogels were prepared on 20 mm coverslips with a solution mixture of 40% acrylamide and 2% bis-acrylamide (Sigma Aldrich, Saint Louis, MO, USA) to generate hydrogels of elastic moduli of 1.10 ± 0.34 and 23.43 kPa [19,20]. Collagen type I from rat tail (0.1 mg/mL) was conjugated to the surface of HG by photo-adhesion with an Irgacure 2959 (0.1 g/mL) (Sigma Aldrich) with Acrylic-*N*-Hydroxysuccinimide-ester (0.02 g/mL) (SCBT), and crosslinking was carried out with a lamp UV light at 350 nm for 3.5 min. Hydrogels were washed with PBS and sterilized with UV light before use.

### 2.7. Statistical Analysis

All statistical analyses were performed using GraphPad Prism 8.0 Software (La Jolla, CA, USA). All data are expressed as the mean +/− standard deviation using unpaired Student’s *t*-test. A *p*-value < 0.05 was considered statistically significant.

## 3. Results

### 3.1. IPF Fibroblasts Have a Decrease in Nuclear HDAC3 Associated with Chromatin Changes

HDAC3 regulates the chromatin structure and its function depends on its nuclear location. To evaluate the levels and cellular localization of HDAC3, we examine fibroblast from IPF (*n* = 3) and control (*n* = 3) by immunofluorescence. Intriguingly, IPF fibroblasts have a higher expression of HDAC3 than control fibroblasts. However, in IPF cells, it was mainly located in the cytoplasm, while control fibroblasts have lower amounts of HDAC3, but predominantly in the nuclei (Figure 1). Measurement of the fluorescence intensity of HDAC3 by colocalization with nuclear stain shows a significant decrease in IPF compared with the control fibroblasts (Figure 1b). As shown in Figure 1c,d, immunoblotting of HDAC3 in the fraction of nuclear proteins corroborates these results.

HDAC3 participates in chromatin condensation, which impacts the nuclear size. As illustrated in Figure 1e, the nuclear area in IPF fibroblast increases significantly (238.9 ± 125.9 µm), and is almost twice the size of the control nuclei (145.1 ± 84.6 µm). Transmission electron microscopy also suggests that the nuclear size in IPF fibroblasts is more prominent than in controls (Appendix A). These results indicate that nuclear HDAC3 is decreased in IPF fibroblasts, which correlates with an increase in nuclear area and could suggest an alteration in chromatin compaction dynamics in IPF.

### 3.2. Nuclear HDAC3 Mislocalization Decreases Heterochromatin Histone Marks and Promotes Profibrotic Gene Expression

Histone acetylation alters the dynamics of compaction between euchromatin and heterochromatin. One of the targets of HDAC3 is the H3K9 residue; heterochromatin formation depends on the deacetylation of this residue, to be methylated later. As shown in Figure 2, IPF fibroblasts have a decreased peripheral chromatin density compared with controls (Figure 2a, yellow arrows), indicating the possible loss of heterochromatin and lamin-associated domains (LADs). To confirm heterochromatin loss in the IPF fibroblasts, H3K9me2, a hallmark of transcriptional silencing, was assessed. Immunofluorescence colocalization of H3K9me2 with lamin A revealed a dramatic loss of H3K9me2 in the nuclear periphery of IPF fibroblasts compared with controls (Figure 2b), and this was confirmed by immunoblot (Figure 2c). In addition, IPF fibroblasts have a significantly higher expression of H3K9ac than controls, which correlates with a decrease in H3K9me2, as shown in Figure 2c. Thus, the lack of colocalization of H3K9me2 with nuclear lamin A observed in IPF fibroblasts is likely associated with a change in the H3K9 acetylation/methylation ratio.

To further confirm the role of HDAC3 in the increased H3K9 acetylation observed in IPF fibroblasts, we treated control fibroblasts with RGFP966, a highly selective HDAC3 activity inhibitor with an IC50 of 80 nM [21]. RGFP966 increased H3K9ac fluorescence intensity in a dose-dependent manner; Figure 3a. As expected, inhibition of HDAC3 by RGFP966 induced a marked increased acetylation and a decrease in methylation; Figure 3b,c. Moreover, control fibroblasts treated with the inhibitor exhibit similar changes in the nuclear area to those seen in IPF fibroblasts (206 ± 87.04 µm^2^), twice the size of non-treated fibroblasts (103.8 ± 49.88 µm^2^); Figure 3d. Importantly, HDAC3 expression or translocation was not affected by inhibitor treatment, indicating that the observed effects are due to activity inhibition (Figure 3b and Appendix A).

Loss of nuclear HDAC3 or its activity led to the hyperacetylation of H3K9, a euchromatin mark associated with active gene transcription, and we wondered whether this might be reflected in the expression of fibrotic-related genes. Therefore, Col1A1, αSMA, TGFβ, and p21 expression were assessed in fibroblasts with inactivated HDAC3 and evaluated by real-time PCR. Normal human lung fibroblasts treated with HDAC3 inhibitor (RGFP966) display an increased expression of Col1A1 and αSMA compared with non-treated fibroblasts, while no differences were found in TGFβ expression Figure 4.

As HDAC3 has a role in the cell cycle, and there is evidence suggesting that IPF fibroblasts present a senescent phenotype, we next evaluated the effect that HDAC3 has on molecules associated with senescence and the cell cycle. Normal lung fibroblasts treated with RGFP966 displayed an increased expression of p21 compared with non-treated fibroblasts (Figure 4).

Therefore, pharmacological inhibition of HDAC3 activity in fibroblasts induces hyperacetylation of H3K9, which is associated with increased transcriptional activity and correlates with an increase in the expression of αSMA, collagen, and p21. Likewise, the inhibition of HDAC3 in control fibroblasts causes a decrease in the expression of genes associated with cell proliferation, such as MKi67, PCNA, and PLK1 (Appendix A).

### 3.3. Extracellular Matrix Stiffness Causes a Decrease in HDAC3, Promoting Fibrosis-Related Gene Expression

Among the processes involved in the progression of fibrosis, the increased accumulation and stiffness of the extracellular matrix result in a sustained remodeling, altering, among others, the nuclear architecture [22]. However, the role of HDAC3 in this process is unknown.

To identify if the matrix stiffness impacts fibroblasts on HDAC3 expression, we grow these cells on polyacrylamide hydrogels that mimic the stiffness of a normal lung (1 kPa) and fibrotic lung (23 kPa). As illustrated in Figure 5, control cells displayed a decrease in the HDAC3 amount in response to matrix stiffness. However, no changes were observed in the immunoreactive protein in IPF fibroblasts, indicating that matrix stiffness does not affect the expression of HDAC3 (Figure 5a). HDAC3 quantification by colocalization with nuclear stain in control fibroblasts shows a significant decrease in HDAC3 in the stiff matrix, but not in IPF fibroblasts (Figure 5b). To further confirm these changes, HDAC3 expression was analyzed by immunoblot; in control cells, there is an inverse correlation between the amount of HDAC3 and stiffness. However, this correlation is virtually not seen in IPF cells (Figure 5c). These results reveal that the decrease in HDAC3 in control fibroblasts may result from the cellular response to increased stiffness. We also wondered if this could impact the expression of the altered genes when the deacetylase inhibitor was used. Then, we also analyzed the expression of Col1A1, αSMA, TGFβ, and p21 in the same cells seeded in hydrogels. As shown in Figure 5d, fibroblasts cultured on a stiff matrix have a higher amount of Col1A1, αSMA, and p21 expression than cells grown in a soft matrix (1 kPa) (Figure 5d). This loss of HDAC3 by mechanical stress (stiffness) has a similar effect on profibrotic gene expression to those seen with the pharmacological inhibition of HDAC3.

Finally, to determine if nuclear HDAC3 expression and localization depend on the mechanosensitive response of cells to ECM through the actin cytoskeleton, we used an inhibitor of actin filaments’ polymerization, latrunculin B (LatB), at a concentration (1 μM) and time that does not have a cytotoxic effect on cells and that, under our conditions, resulted in 90% viability analyzed by calcein-AM and propidium iodide staining (data not shown). As illustrated in Figure 6a, the perturbation of actomyosin filaments resulted in changes in HDAC3 expression. The results show that normal lung fibroblasts treated with latrunculin B have an increased expression of HDAC3, which was corroborated by immunoblot (Figure 6b), and fractionation of cell lysates indicates that this occurs mainly in the nucleus of fibroblasts (Figure 6c). Interestingly, we also found that IPF fibroblasts respond to LatB treatment and observed that a portion of HDAC3 translocated to the nucleus (Figure 6a).

## 4. Discussion

HDACs are responsible for maintaining a compact chromatin and preventing gene expression by removing these acetyl groups [23]. Despite the studies carried out trying to elucidate the role that HDACs have in the pathogenesis of IPF, it is still unknown. Therefore, the aim of this project was to determine the influence of HDAC3 on chromatin remodeling and gene expression associated with idiopathic pulmonary fibrosis. Our results show that HDAC3 is decreased in the nucleus of IPF fibroblasts, favoring chromatin opening. A lack of nuclear HDAC3 causes a hyperacetylation and a decrease in the heterochromatin mark (H3K9me2) and the expression of the profibrotic profile. Moreover, the mechanical stimulus of matrix stiffness present in the IPF ECM decreases the expression of HDAC3 on normal lung fibroblasts, which consequently increases the expression of Col1A1, αSMA, and p21 (Figure 7).

Notably, in IPF, there is very little information regarding HDAC3. A strong nuclear expression in basal cells and type II alveolar epithelial cells and an increase in fibroblasts have been previously reported [9,24]. Nevertheless, those studies ignore the subcellular location of HDAC3 in fibroblasts, which is crucial for it to carry out its function. Regardless of being overexpressed, we demonstrated for the first time that HDAC3 is found outside the nucleus of IPF-derived fibroblasts. Therefore, it cannot execute its function of deacetylating histones. Although the regulation of HDAC3 is not known in detail, there is evidence that interaction with complex SMART/NCOR1 is required to engage its catalytic activity and maintain its nuclear location [25,26]. Therefore, dissociation of the corepressor complex makes these proteins susceptible to degradation in the cytoplasm [27]. In the case of IPF, we speculated that this accumulation in the cytoplasm could be associated with a decrease in degradation pathways without discarding specific export and may signify that a non-histone target remains unknown [28,29]. The lack of nuclear HDAC3 activity in IPF fibroblasts may have substantial implications for disease development and progression because, in other cellular models, HDAC3 participates in establishing the genetic profile necessary for cell proliferation, migration, apoptosis, autophagy, and senescence; all of these processes are also related to IPF [30,31,32,33].

One feature that is important to note about IPF fibroblasts is nuclear size. As shown, nuclear size differs between normal and IPF-derived fibroblasts, with the last being twice the size. Studies carried out in cancer have shown that the increase in the size of the nucleus is a hallmark of cancer cells, and it is considered in some types of cancer as a diagnostic and prognostic marker [34,35,36]. Interestingly, tumor regression in response to cancer treatment is associated with decreased nuclear size [37]. In IPF, nuclear size should be a point of study because of the impact that the change in nuclear morphology has on the expression signature.

Nuclear size and shape may be directly affected by changes in histone modifications modulated by HDAC3, where cells showing higher nuclear HDAC3 expression have smaller nuclei [38,39,40]. Histone acetylation has long been recognized to change nuclear size by promoting a looser structure and transcriptionally active chromatin [41,42]. Although the factors that influence nuclear morphology are known, the mechanisms and molecules involved remain to be determined. Here, we hypothesize that HDAC3 is crucial for maintaining nuclear architecture in lung fibroblasts and that alterations in the expression and localization of this protein are closely related to changes in chromatin compaction and genomic stability. An unveiling result, demonstrated by transmission electron microscopy, is that IPF-derived fibroblasts display a decreased amount of heterochromatin, mainly in the periphery. These results are consistent with the lack of nuclear HDAC3 shown on IPF fibroblasts because its deletion is associated with an overall decrease in heterochromatin and lamina-associated domains (LADs) [16,17,40,43,44].

HDAC3 is required for maintaining the acetylation/methylation ratio at H3K9 residue [29,45,46]. In the context of IPF, it has been shown that there is hyperacetylation. Consequently, as demonstrated by Hanmandlu et al., alterations in chromatin accessibility led to gene expression of fibrotic-related pathways [47]. In agreement with these findings, we discovered that IPF fibroblasts display a higher amount of H3K9 acetylation, and inhibiting its activity as a deacetylase in normal lung fibroblasts causes a defect in the H3K9ac/H3K9me2 ratio due to a hyperacetylation. These results suggest that IPF fibroblasts display a transcriptionally active chromatin pathway through the lack of nuclear HDAC3.

To determine if these chromatin changes are associated with the development or progression of the fibrotic process, we evaluated the expression of some genes strongly associated with IPF. We found that, when we brought down HDAC3 deacetylase activity, there was an elevated expression of the α-smooth muscle actin, suggesting a fibroblast to myofibroblast differentiation, as well as of the extracellular matrix component collagen I. Importantly, we found that TGFβ expression does not change if HDAC3 activity is abolished. Although it was expected to change, this result is consistent with other studies where it has been shown that pharmacological inhibition of HDAC3 can induce cell differentiation without significant changes in TGFβ levels [48]. Likewise, IPF is an age-dependent disease and senescent fibroblasts have been identified in the lungs of patients with IPF [49]. Therefore, we assessed p21 as a senescence marker, which we found to be increased in the absence of HDAC3 activity. These results are in agreement with those reported in the literature, where it is proved that acetylation regulation is a crucial mechanism in lung fibrosis development [50]. Our results indicate that HDAC3 regulates transcriptional repression of several profibrotic genes and that its deficiency enhances the fibrotic response. Nevertheless, the evidence of the expression, activity, and function of this enzyme in the context of IPF is not yet known with precision, and further studies will be needed.

Lung fibrosis is characterized by an increase in matrix stiffness due to ECM accumulation, which establishes a profibrotic loop [18,51]. In this sense, HDAC3 has been reported to participate in the mechanisms of mechano-transduction response, where matrix stiffness or geometric cell constraints drove changes in the nucleus morphology, chromatin accessibility, hyperacetylation, and gene expression [39,52,53]. In agreement with what was reported, our findings show that there is an inversely proportional correlation between matrix stiffness and the expression of HDAC3 in normal human lung fibroblasts, allowing the transcription of profibrotic genes. It is important to highlight that biophysical cues, such as increased matrix stiffness, directly affect chromatin remodeling and that epigenetic mechanisms, such as microRNAs and histone modifications, integrate these signals and respond by restructuring chromatin accessibility [54,55,56].

In the same line, LatB was used to demonstrate that HDAC3 regulation depends on actomyosin contractility. We found that depolymerization of actin filaments with LatB increases nuclear HDAC3 expression in normal fibroblasts. Interestingly, we also observed this phenomenon in IPF fibroblasts, confirming that HDAC3 responds to mechanotransduction signals. Our results agree with those published by Jain et al., who demonstrated that disruption of actin filaments by cytochalasin b or LatB, both actin depolymerization drugs, resulted in cytoplasmic to nuclear translocation of HDAC3 and a decrease in H3K9 acetylation mark. Future experiments are needed to establish whether chemical actin depolymerization has the same effect on the fibrosing phenotype of these cells [39]. Accordingly, our findings demonstrated that HDAC3 is a mechanoresponsive molecule that plays an essential part in the signaling axis between stiffness and epigenetic remodeling displayed in IPF fibroblasts, suggesting that control of myofibroblast activation by stiffness is associated with chromatin remodeling by HDAC3.

In summary, these results demonstrate that loss of nuclear HDAC3 by pharmacological inhibition or induced by stiffness has a profibrotic effect on human lung fibroblasts. Furthermore, HDAC3 could be the link between microenvironmental signaling and epigenetic response in the pathogenesis of IPF.

## Figures and Tables

**Figure 1 cells-12-01020-f001:**
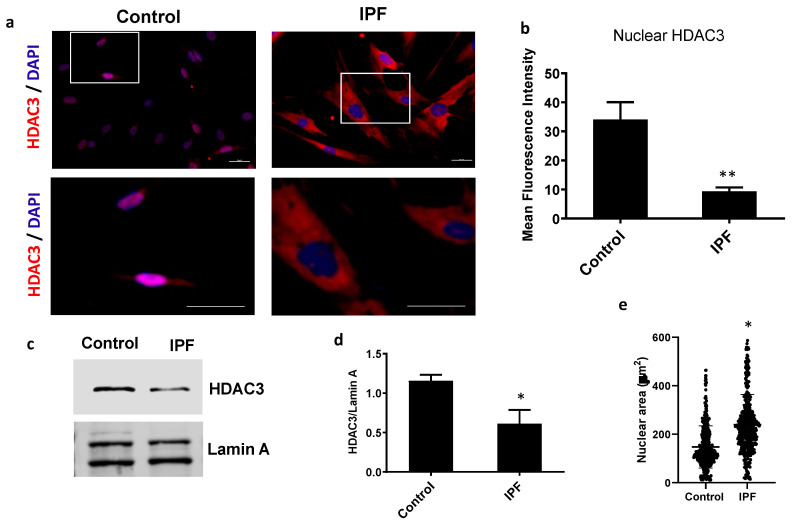
Nuclear HDAC3 decrease in IPF fibroblasts. (**a**) Representative image of HDAC3 localization (red) by immunofluorescence in control and IPF cells stained with DAPI for nuclei (blue); scale bar: 20 μm. The bottom images are a close-up of the box shown in the upper. (**b**) Bar graph represents nuclear HDAC3 quantification of three control and three IPF cell lines (mean ± SD). (**c**) Representative Western blot of HDAC3 on nuclear extracts from control and IPF fibroblasts. Lamin A/C was used as a loading control. (**d**) Densitometric analysis of Western blot of three controls and three IPF cell lines shows data as mean ± SD. (**e**) Graph of the nuclear-projected area of three different cell lines per condition (mean ± SD) of two independent experiments. * *p* < 0.05; ** *p* < 0.01 by Student’s *t*-test.

**Figure 2 cells-12-01020-f002:**
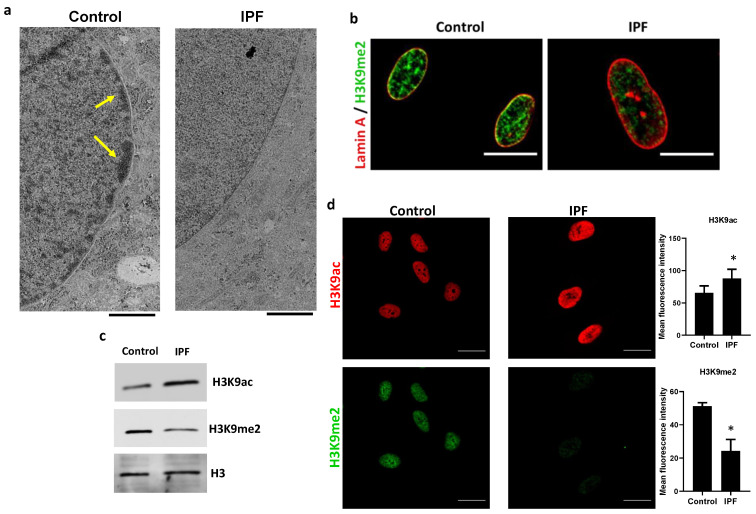
Reduction in peripheral heterochromatin and increase in acetylation mark (H3K9ac) in IPF fibroblasts. (**a**) Representative microphotographs of transmission electron microscopy of nuclear peripheral heterochromatin associated with lamina indicated with yellow arrows; scale bar = 1 μm. (**b**) Colocalization of H3K9me2 (heterochromatin histone mark) with lamin A by confocal microscopy; scale bar = 20 μm. (**c**) Representative immunoblot of control and IPF fibroblasts with H3K9ac and H3K9me2; H3 total was used as loading control. (**d**) Representative immunofluorescence images of control and IPF fibroblasts with H3K9me2 (green) and H3K9ac (red); scale bar = 50 μm. Graphs show fluorescence intensity quantification; each bar represents the mean ± SD from three control and three IPF cell lines of two independent experiments. * *p* < 0.05; by Student’s *t*-test.

**Figure 3 cells-12-01020-f003:**
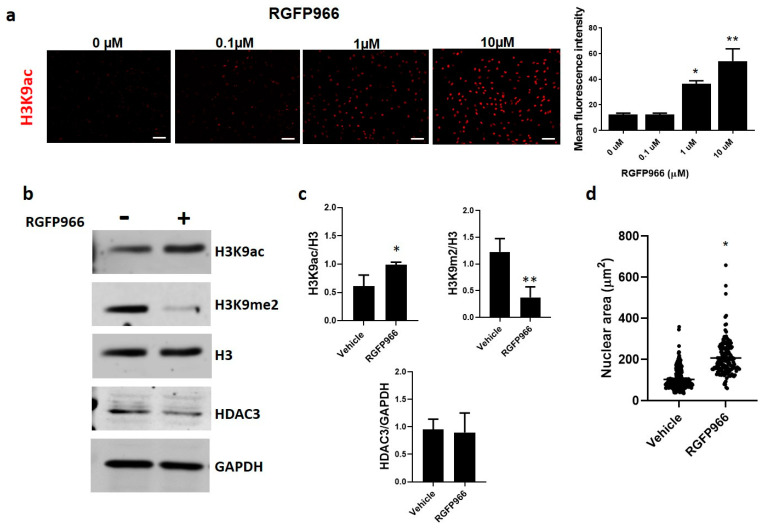
Pharmacological inhibition of HDAC3 by RGFP966 favors H3K9 acetylation, decreases H3K9 methylation, and increases nuclear area. (**a**) H3K9 acetylation of control fibroblasts treated with different concentrations of RGFP966 for 24 h; scale bar 100 µm. The graph represents the quantification of fluorescence intensity; each bar represents mean ± SD from three independent experiments. (**b**) Representative immunoblot of H3K9ac, H3K9me2, and HDAC3 from fibroblasts treated with RGFP966 10 µM for 24 h. H3 and GAPDH were used as loading controls. (**c**) Densitometric analysis of immunoblots from three independent experiments. (**d**) A measure of nuclear-projected area upon control and treated cells (RGFP966 10 µM); data are given as mean ± SD from three independent experiments. * *p* < 0.05; ** *p* < 0.01 by Student’s *t*-test.

**Figure 4 cells-12-01020-f004:**
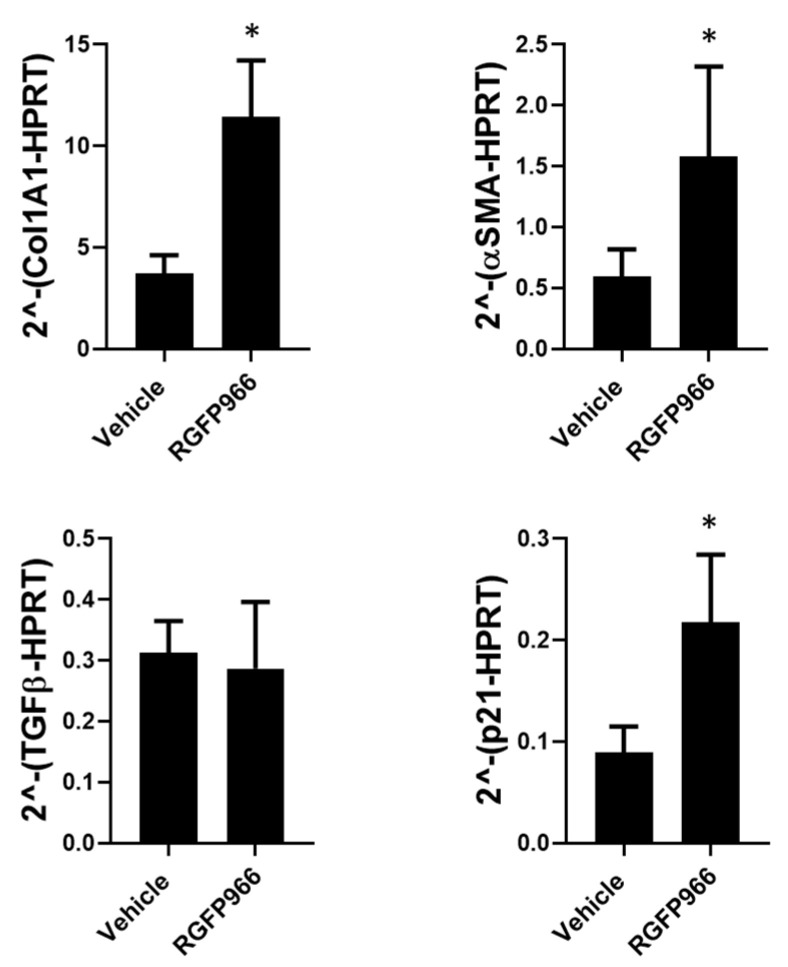
Pharmacological inhibition of HDAC3 by RGFP966 regulates the expression of profibrotic genes. qPCR analysis of fibrosis-related genes: Col1A1, αSMA, TGFβ, and p21; control fibroblasts treated with RGFP966 10 µM for 24 h from three independent experiments. * *p* < 0.05 by unpaired Student’s *t*-test.

**Figure 5 cells-12-01020-f005:**
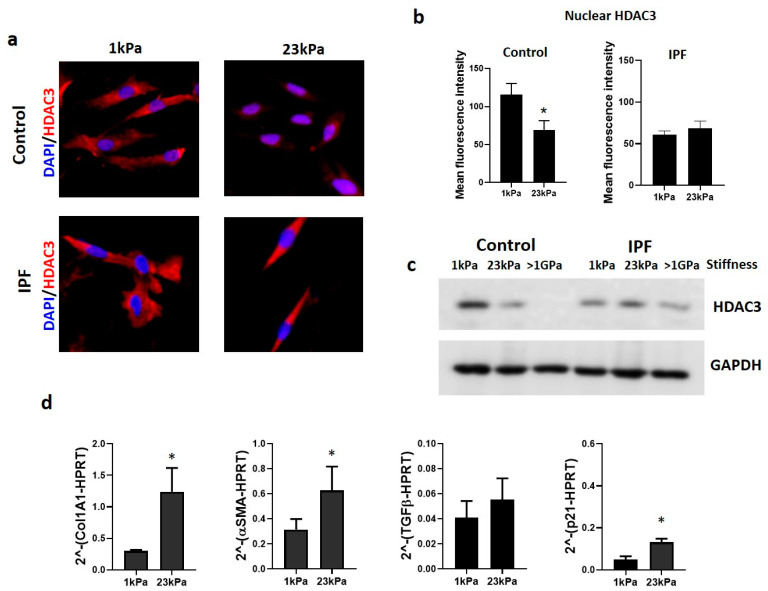
Matrix stiffness causes a decrease in HDAC3 in control fibroblasts and regulates the expression of profibrotic genes. (**a**) Representative immunostaining images of HDAC3 in control and IPF fibroblasts cultured on polyacrylamide hydrogels of 1 and 23 kPa; HDAC3 (red) and nuclei (blue). (**b**) A measure of mean fluorescence intensity of nuclear HDAC3 from three independent experiments; data presented as mean ± SD. * *p* < 0.05 by unpaired Student’s *t*-test. (**c**) Representative immunoblot of HDAC3 from fibroblast-cultured polyacrylamide hydrogels of 1, 23 kPa, and <1 GPa (culture plates) for five days. (**d**) qPCR analysis of fibrosis-related genes, Col1A1, αSMA, TGFβ, and p21, which increase expression when cells are cultured on 23 kPa hydrogels. * *p* < 0.05; by unpaired Student’s *t*-test.

**Figure 6 cells-12-01020-f006:**
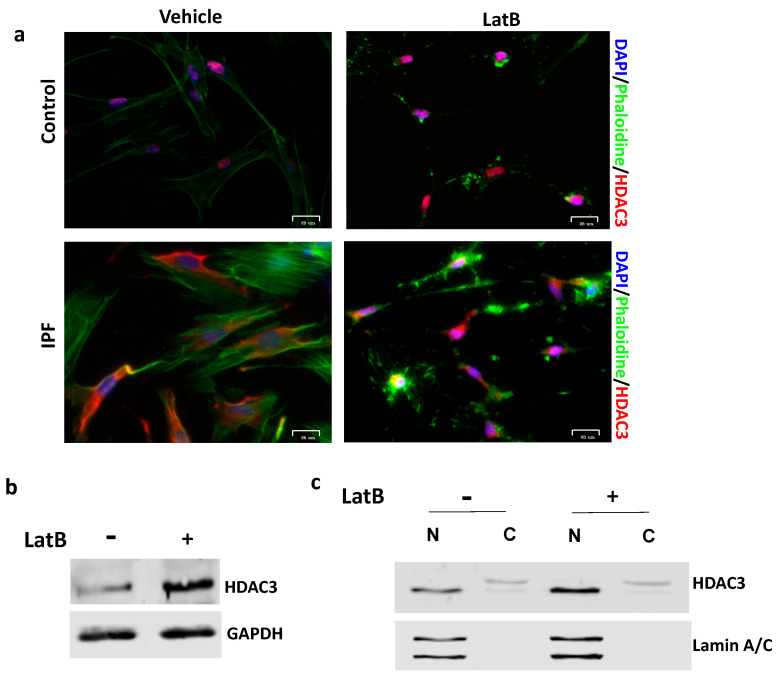
The actin cytoskeleton regulates the expression and localization of HDAC3. (**a**) Immunofluorescence images of HDAC3 in control and IPF fibroblasts non-treated and treated with latrunculin B (1 µM) for 70 min, HDAC3 (red), phalloidin (green), and DAPI (blue). Representative immunoblots of HDAC3 from (**b**) the total lysate and (**c**) the nuclear and cyoplasmic extracts of fibroblasts treated or not with the actin polymerization inhibitor.

**Figure 7 cells-12-01020-f007:**
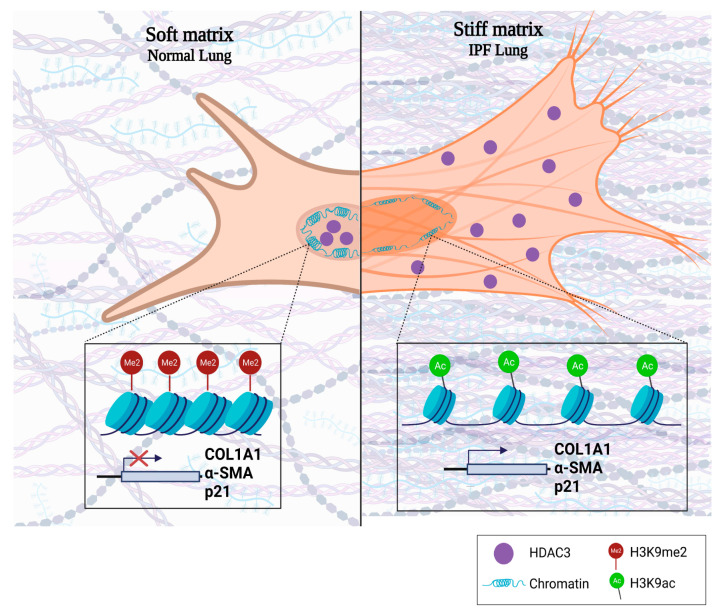
Created with BioRender.com. Matrix stiffness triggers the absence of HDAC3, chromatin remodeling, and fibroblast activation. HDAC3 is a mechanoresponsive protein crucial for maintaining compact chromatin and preventing gene expression by removing acetyl groups and increasing histone methylation. (**Left**) HDAC3 localizes mainly to the nucleus in a soft matrix and promotes peripheral compaction through its deacetylase function. (**Right**) In contrast, HDAC3 is exported to the cytoplasm in a rigid matrix, promoting hyperacetylation, more open chromatin, increased nucleus size, gene expression, and fibroblast activation.

## Data Availability

All data are contained within the article.

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
