# Peer review of "Absence of HDAC3 by Matrix Stiffness Promotes Chromatin Remodeling and Fibroblast Activation in Idiopathic Pulmonary Fibrosis"

_cells, 2023, doi:10.3390/cells12071020_

Round 1

Reviewer 1 Report

The paper "Absence of HDAC3 by matrix stiffness promotes chromatin remodeling and fibroblast activation in Idiopathic Pulmonary Fibrosis" by Toscano-Marcos and collaborators show the correlation between the loss of nuclear HDAC3 with loss of heterochromatin and the expression of pro-fibrotic and senescence markers in normal lung fibroblasts, as well as the effect of mechanical changes in the extracellular matrix stiffness of the localization of HDAC3 and the effect of HDAC3 localization in chromatin condensation and epigenetic regulation of gene expression.  

The paper is well written with adequate experiments, however it presents major experimental flaws that need to be addressed before the work can be published:

Major Concerns

1. What is the level of nuclear translocation of HDAC3 in normal fibroblasts that are treated with  RGFP966? 

2. What is the effect of Latrunculin B in IPF fibroblasts? and what is the effect of Latrunculin B in cells that are cultured on a stiff matrix? If translocation responds to mechanical stress, cells treated with the inhibitor should not respond to changes in the extracellular matrix stiffness.

Minor

typo. Line 192 RGFP66.

Please provide a reference for the efficiency and specificity of RGFP966 

Please include in your discussion how HDAC3 is regulated, and the difference between normal/IPF fibroblasts, why HDAC3 in IPF fibroblasts have cytoplasmic localization.

The paper would benefit if a visual representation of the model was included as a figure or as a visual abstract in the paper.

Reviewer 2 Report

This is an interesting and possibly important manuscript that describes the role of the epigenetic protein and enzyme HDAC3.  HDAC3 is responsible for decatylating histones associated with chromatin and therby are classical epigenetic regulators of gene expression and are involved in fibrosis.  They first show that nuclear only HDAC3 is reduced in idiopathic pulmonary fibroblasts by ICC and western immunoblotting. This is convincing but can the authors not show a better immunoblot than that shown?  The Lamin A loading control looks off?? Must have a better blot than that to illustrate the changes.

They then subsequently show reduced heterochromatin and increased HK39ac acetylation mark in these cells. The two are interlinked and one would expect this. They use immunofluorescence and microscopy to illustrate this. This is interesting approach but must be backed up with an Western Immunoblot to confirm this in the IPF cells.

Figure 4 they show the gene expression of TGF-beta1fter RGFP966a treatment to inhibit HDAC3. They demonstrate no significant changes to the gene expression but this may not neccesarily reflect the true change of TGF-beta1 at the protein levels and its pretty easy to measure TGF-beta1 by a commercial ELISA i.e. R&D systems one. I feel this is critically important as TGF-beta1 would be expected to change in this situation.

Lat B is used to inhibit cellular actin polymerisation and show that treatment with Lat B resulted in promotion of HDAC3 nuclear translocation thus actin is involved in its movement and localisation. Lat B can be toxic; can the authors confirm that there was no significant toxicity in the treated cells? Toxicity would also alter nuclear cyto ratio.
Do you see the same effect if you treat with Cytochalasin B?  This would co-berate your findings.

Minor:
Reference 33 should include reference Tsou PS et al Nature Reviews Rheumatology 2023 17: 596-607.

Round 2

Reviewer 1 Report

I consider that the authors responded to all my concerns and I am happy to recommend the manuscript for publication.